# Clinical Relevance of Drug Interactions with Cannabis: A Systematic Review

**DOI:** 10.3390/jcm11051154

**Published:** 2022-02-22

**Authors:** Valentina Lopera, Adriana Rodríguez, Pedro Amariles

**Affiliations:** Research Group on Pharmaceutical Promotion and Prevention, Department of Pharmacy, University of Antioquia, UdeA, AA 1226, Medellin 050010, Colombia; valentina.loperag@udea.edu.co (V.L.); arodriguezbedoya@gmail.com (A.R.)

**Keywords:** drug interactions, medicinal cannabis, recreational cannabis, clinical relevance, severity, probability

## Abstract

Concomitant use of cannabis with other drugs may lead to cannabis–drug interactions, mainly due to the pharmacokinetic mechanism involving the family of CYP450 isoenzymes. This narrative systematic review aimed to systematize the available information regarding clinical relevance of cannabis–drug interactions. We utilized the PubMed/Medline database for this systematic review, using the terms drug interactions and cannabis, between June 2011 and June 2021. Articles with cannabis–drug interactions in humans, in English or Spanish, with full-text access were selected. Two researchers evaluated the article’s inclusion. The level of clinical relevance was determined according to the severity and probability of the interaction. Ninety-five articles were identified and twenty-six were included. Overall, 19 pairs of drug interactions with medicinal or recreational cannabis were identified in humans. According to severity and probability, 1, 2, 12, and 4 pairs of cannabis–drug interactions were classified at levels 1 (very high risk), 2 (high risk), 3 (medium risk), and 5 (without risk), respectively. Cannabis–warfarin was classified at level 1, and cannabis–buprenorphine and tacrolimus at level 2. This review provides evidence for both the low probability of the occurrence of clinically relevant drug interactions and the lack of evidence regarding cannabis–drug interactions.

## 1. Introduction

Nowadays, medical cannabis is considered a promising option for the treatment of certain diseases, for instance, epilepsy and chronic pain [1,2]. In addition, some authors have proposed it as a possible treatment for COVID-19 due to its anti-inflammatory properties [3]. However, the evidence is not robust for COVID-19 nor for other inflammatory diseases, such as Crohn’s disease and ulcerative colitis [4]. Throughout history, cannabis has been recognized for its medical uses, being included in the U.S. Pharmacopoeia in 1850, although it was later excluded [5]. Since 1990, scientific articles related to cannabis, marijuana, cannabinoids, tetrahydrocannabinol, and cannabidiol have increased, showing a great interest in its study and a change of attitudes towards the use of this plant [6]. A survey conducted in the United States showed how the use of this substance among older adults has increased [7]. Three cannabinoid-containing medicinal products are currently on the market: dronabinol, which is a synthetically obtained delta-9-tetrahydrocannabinol (THC); Sativex, which is a combination of THC and cannabidiol (CBD); and Epidiolex, which is an oral solution of CBD.

A drug interaction corresponds to a modification, which is quantifiable in the magnitude or duration of effects related to the simultaneous or previous administration of other drugs, food, or to the pathophysiologic conditions of the patient [8]. It can be additive (1 + 1 = 2), synergistic (1 + 1 > 2), or antagonistic (1 + 1 < 2). Generally, most of the interactions reported in the literature are explained by a pharmacokinetic mechanism, mediated by an alteration in the activity of the isoenzymes of the family cytochrome P450 enzyme (CYP), the glycoprotein P (P-gp), or by other drug transporters. Cannabinoids, especially CBD, are metabolized by isoenzymes of the family CYP, which can lead to interactions between a cannabis base extract and other drugs. In this way, recent in vitro studies have shown that: THC competitively inhibits the enzymes CYP1A2, CYP2B6, CYP2C9, and CYP2D6; CBD competitively inhibits the enzymes CYP3A4, CYP2B6, CYP2C9, CYP2D6, and CYP2E1; and cannabinol (CBN) competitively inhibits the enzymes CYP2B6, CYP2C9, and CYP2E1 [9]. By contrast, drug interactions due to pharmacodynamic mechanisms are generally less common [6].

Currently, cannabis–drug interactions are mostly theoretical or come from case reports due to the lack of clinical trial results that evidences the effects of the interactions and the probability of their occurrence. Additionally, the different routes of administration and their recreational use limit the comparability and standardization of the information [6]. Therefore, the information related to the clinical relevance of cannabis–drug interactions is limited, and, as such, it is necessary to structure and systematize the available information about this topic. In this context, the aim of this study is to systematize the available information about the clinical relevance of cannabis–drug interactions, contributing to the safe medicinal use of these kinds of products. 

## 2. Materials and Methods

We utilized the PubMed/Medline database for this systematic review, using the following as a search strategy: (Drug interactions [Title/Abstract]) AND (Cannabis [Title/Abstract]), with filters for articles published between 06/2001 and 06/2021, in English or Spanish, with full-text access. 

Inclusion criteria: All types of articles with relevant clinical information regarding drug interactions in humans either using cannabis to treat diseases or for recreational purposes were included. 

Exclusion criteria: (a) Preclinical studies (in animals or in vitro); (b) articles with information without the support of additional studies (theoretical interactions); (c) articles without specific information about cannabis—drug interactions; and (d) articles without full-text availability.

The studies identified were reviewed by two researchers in pairs according to the preferred reporting items for systematic reviews and meta-analysis (PRISMA) flow chart via pre-determined eligibility criteria. Thus, titles and abstracts of all identified articles were screened for eligibility. Both authors analyzed articles selected and their inclusion was defined by consensus. Relevant references of the reviewed articles were also included.

Subsequently, the information was registered in a database according to the following items: article name, cannabis product evaluated, drug-related interaction, clinical relevance level (according to the combination of severity and probability of occurrence), pharmacodynamic or pharmacokinetic mechanism, comment, recommendation, and reference. The data registered were proofread by another author.

Pairs of identified cannabis-drug interactions were classified into five levels, according to severity (effect on patient’s health) and probability of occurrence (type of study that supports the interaction), following the combination of options as shown in Table 1 [8,10]. 

The probability was determined and classified according to type of study that supported the interaction found for each pair of cannabis–drug interactions: Possible: The interaction was documented by results from less than three case reports.Probable: The interaction was documented by results from at least one observational study (cohort or case–control study) or a least three case reports.Defined: The interaction was documented by results from at least one meta-analysis, narrative systematic review, or randomized or non-randomized clinical trial.

The severity was determined according to the magnitude of the qualitative change in parameters associated with the efficacy or safety of the drug, such as pharmacokinetics (clearance or area under the curve (AUC)) or clinical (aspartate aminotransferase (AST), alanine aminotransferase (ALT), or international normalized ratio (INR) levels). Thus, the alteration was assessed with the data available before and after the interaction. The calculation performed to assess the changes in the parameter related to the safety and/or efficacy of the drugs was: $$\frac{\left(Par\;int-Par\right)}{Par}*100=$$
where:

Par in: Parameter during the interaction (AUC, clearance, AST, ALT, or INR).

Par: Parameter before the interaction or after the interaction, or the average of the normal concentration. 

Then, the severity was established according to the magnitude of the parameter alteration [11].

Minor (the interaction does not cause or causes minimum harm to the patient): The variation of the parameter was between 25% and 100% (INR, AST, ALT, or AUC), or between 20% and 50% (clearance).

Moderate (the interaction generates the need for closer monitoring of patient health): The variation of the parameter was between 100% and 400% (INR, AST, ALT, or AUC), or between 50% and 80% (clearance).

Severe (the interaction can cause harm or injury to the patient): The variation of the parameter was 400% or more (INR, AST, ALT, or AUC), or 80% or more (clearance).

## 3. Results

### 3.1. Results of the Search

In the PubMed/Medline database, 94 articles were identified. The titles and abstracts of 93 of these were screened, leading to the selection of 53 eligible articles that were read in full text. Based on our predefined exclusion criteria, 27 articles were further excluded; therefore, 26 articles were included in the review (Figure 1).

### 3.2. Drug Interactions 

Nineteen pairs of drug interactions with medicinal or recreational cannabis were identified, with drugs of four pharmacological groups, as shown in Table 2.

The level of clinical relevance of cannabis–drug interactions and comments and recommendations are shown in Table 3. According to severity and probability, the 19 pairs of cannabis–drug interactions were classified at level 1 (one pair), level 2 (two pairs), level 3 (12 pairs), or level 5 (four pairs).

## 4. Discussion

### 4.1. Evidence

Cannabis–drug interactions mainly occur in cases where cannabinoids are metabolized or alter the metabolic activity of some isoenzymes of the CYP family. Therefore, simultaneous administration may induce or inhibit the metabolic activity of cannabis or any concomitant drug metabolized by the same isoenzymes of the CYP family. The identification and prevention/control of these variations may contribute to the safe use of medicinal cannabis.

This narrative systematic review identified and classified the level of clinical relevance for 19 pairs of cannabis–drug interactions (one corresponding to level 1, two to level 2, 12 to level 3, and four to level 5), mainly when cannabis is used for medical purposes. These findings are scientifically supported for prescription and dispensation processes and contribute to increasing the probability of achieving the therapeutic objectives of cannabis use for some specific illnesses in health systems. In addition, healthcare professionals may use this information to advise patients about the possible effects of recreational cannabis use in the results of pharmacotherapy. 

The cannabis–warfarin interaction was classified at level 1 of clinical relevance (high risk). Although only four case reports were found as evidence, in all cases, the interaction led the patient to consult for bleeding and the INR level increased to 7.6 on average. In addition, in a review conducted by Ge et al. [41], herb–warfarin interactions were classified by severity and level of evidence. Severity was classified as major, moderate, minor, and nonclinical, and levels of evidence as highly probable, probable, possible, and doubtful. Similar to the results from this review, the interaction between *Cannabis Sativa* L. and warfarin was classified as major and possible. In addition, Greger et al. [42], in a review of warfarin interactions, found in vivo evidence (case reports) regarding cannabis–warfarin interactions, and theorized similar interactions, for example, with clopidogrel, based on the isoenzymes that metabolize this drug and cannabis. 

This increase in the INR levels is explained by a pharmacokinetic mechanism (inhibition of the metabolic enzymes of warfarin, mainly by CBD). In this way, authors have suggested a correlation between CBD dose and the increase in INR [14]. Therefore, for patients receiving treatment with warfarin and for whom, due to other medical conditions, medical cannabis is indicated, dose adjustment is required to maintain therapeutic INR levels. In addition, patients receiving treatment with warfarin must be advised not to use recreational cannabis. 

Two pairs of interactions, regarding buprenorphine and tacrolimus, were classified at level 2 of clinical relevance. Although none of these interactions is classified as defined due to a lack of clinical trials, the reports and findings indicate an increase in the plasma levels of these drugs. In both cases, controlled clinical trials are required to improve the information about these interactions. The recommendation is to adjust the dosage and monitor the plasma level of the drug, and if it is necessary, the usage of medicinal cannabis must start progressively and under the supervision of a healthcare professional.

Most pairs of drug interactions (12 out of 19) were classified at level 3 of clinical relevance due to weak to moderate changes in pharmacokinetic or clinical parameters. However, this result shows that the usage of medical cannabis requires medical assessment and follow-up favoring the appropriate definition of the dose, content (whether it is a pure extract, a mixture of cannabinoids, or recreational cannabis), and route of administration.

Four pairs of cannabis–drug interactions were classified at level 5 of clinical relevance (interactions with evidence of lack of severity/risk in patient´s health). This information is relevant for clinicians, because it identifies the combination of drugs for which there is evidence that there are no risks to patients from drug interactions [10]. 

Most of the pairs of cannabis–drug interactions (13 of 19) were related to nervous system drugs, mainly anticonvulsants (Table 2). This result may be due to the fact that as an anticonvulsant for rare epileptic disorders, such as Lennox–Gastaut Syndrome and Dravet Syndrome [22], CBD is a therapeutic option that would motivate research for interactions between cannabis and conventional antiepileptic drugs. However, more studies are required to generate information about the usage and results of cannabis–drug interactions, favoring the inclusion of this treatment in guidelines for the management of health problems [26]. 

Regarding pharmacology mechanisms, most of the cannabis–drug interactions were mediated by a pharmacokinetic mechanism. Nonetheless, in the case of interactions with opioids, the mechanism is unknown. On this point, some studies report that vaporized cannabis increases the analgesic effect of morphine and oxycodone [43], without differences in the mean plasma concentration–time curves. As consequence, authors suggest the increase in the analgesic effect is due to a pharmacodynamic mechanism without rejecting the pharmacokinetic mechanism and is based on documentation that CBD and THC enhance the uptake of drugs in animals’ brains [44].

In general, the evidence on cannabis–drug interactions continues to be limited. However, for some drugs, this subject is more precise. For example, for clobazam and CBD, we found three clinical trials representing cannabis–drug interactions with more evidence in this review. Similarly, one review oriented to systematize and update CBD pharmacology in the context of refractory epilepsy found that the strongest evidence is for cannabis–clobazam interactions [45]. In contrast, there are many theorized cannabis–drug interactions or studies with limited clinical evidence, for instance, the interaction between cannabis products (for smoking or oils) in patients treated with nivolumab. In a retrospective study, the use of cannabis was associated with a minor response rate in patients in immunotherapy treatment with nivolumab. However, the magnitude of change in the response rate was not related to cannabis composition. Therefore, to determine the mechanism and clinical relevance of this interaction, it is necessary to conduct clinical trials and generate more information and evidence of cannabis–nivolumab interaction [46]. 

Seven pairs of interactions (37%) were due to recreational use of cannabis, including one with warfarin. Thus, it is important to note that the effect of drug interactions in the health of a patient using recreational cannabis can have a wide range depending on the ingestion route, because it affects the type and quantity of cannabinoids absorbed. In addition, it is necessary to recognize the effect that cigarettes can have on cannabis–drug interactions. For example, in cannabis smokers, chlorpromazine had a 50% increase in clearance, but in the case of cannabis and cigarette smokers, the increase was 107% [25].

The results of this narrative systematic review generate information about cannabis–drug interactions based on the available data. Furthermore, in the cases where there was no explicit information of changes in pharmacokinetic and clinical parameters, quantitative changes were established with the information available in the relevant publication [16,17,19,20,21,22,32,38,39,40], allowing for an extension of the option to classify the clinical relevance of cannabis–drug interactions. 

To improve the rational use of medicinal cannabis, more research is necessary in order to: (a) generate information regarding drug interactions; (b) determinate the type and concentrations of cannabinoids in the respective dosage forms; and (c) improve the standardization of quantities and dosage recommendations.

### 4.2. Limitations of this Review

The results of this review may have some limitations; therefore, the results should be interpreted and used with caution. In this context, the main limitation is the search restriction to a single database, which may not identify other clinically relevant interactions. However, this situation could be minimized with the inclusion of publications identified as relevant in the references of the articles included. In addition, some studies did not describe the cannabis concentration or cannabinoid type used; therefore, as with the case of recreational cannabis, the effect of the cannabis–drug interaction on the patient’s health can have a wide range due to the type and quantity of cannabinoids absorbed. 

## 5. Conclusions

The increased use of medicinal cannabis in the population allows for the development of studies providing the community with scientific supported information that can help to make decisions. This review found and established the clinical relevance of 19 pairs of cannabis–drug interactions, mostly at level 3 of clinical relevance, 1 interaction at level 1 (with warfarin), and 2 interactions at level 2 (with buprenorphine and tacrolimus). These interactions are mediated by a pharmacokinetic mechanism, and most of them are related to nervous system drugs. It is important to emphasize that the confirmation of these findings requires medical assessment and follow-up of on dosage, concentration, cannabis preparation used, and route of administration. Nevertheless, the information found is limited, and it is necessary to conduct clinical trials and to improve the evidence of the effects of cannabis–drug interactions on patients’ health.

## Figures and Tables

**Figure 1 jcm-11-01154-f001:**
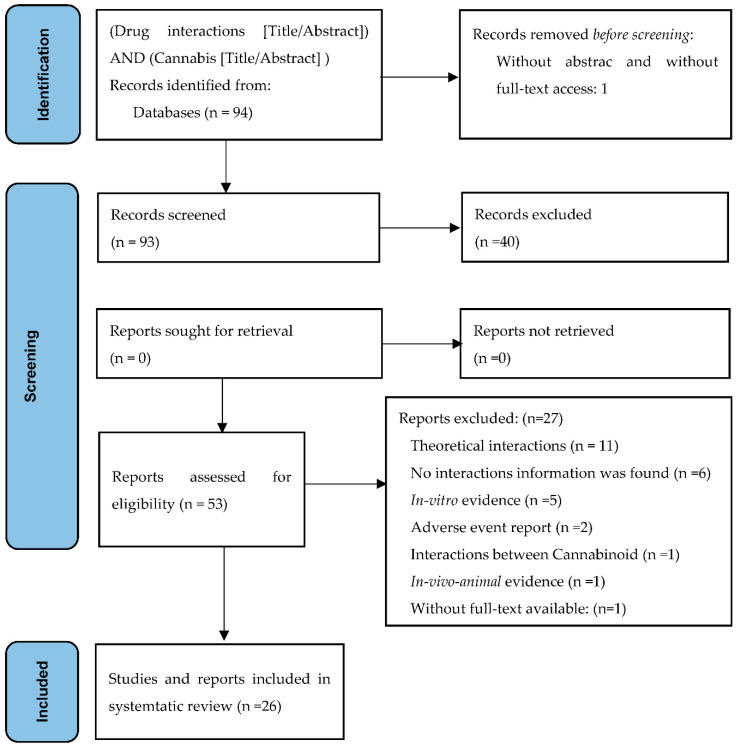
Preferred reporting items for systematic review and meta-analysis (PRISMA): flow diagram for the systematic review of cannabis–drug interactions.

**Table 1 jcm-11-01154-t001:** Levels of the clinical relevance of drug interactions according to the combination of severity and probability of occurrence [8,10].

Severity	Probability		
	Defined	Probable	Possible
Severe	1 (***very high risk***)	1 (***very high risk***)	2 (***high risk***)
Moderate	2 (***high risk***)	2 (***high risk***)	3 (medium risk)
Minor	3 (medium risk)	3 (medium risk)	4 (low risk)
Lack of severity	5 (riskless)	N/A	N/A

Bold and italics indicate the drug interactions more clinical relevant. N/A: No applicable.

**Table 2 jcm-11-01154-t002:** Pharmacological group and level of relevance for 19 pairs of cannabis–drug interactions identified.

Pharmacological Group	Number of Interactions	Level of Relevance	Number by the Level of Relevance (%)
Nervous system	13	2	1 (8)
		3	9 (69)
		5	3 (23)
Blood and hematopoietic organs	1	1	1 (100)
Anti-infectives for systemic use	4	3	3 (75)
		5	1 (25)
antineoplastic agents and immunomodulators	1	2	1 (100)
Total of pairs of cannabis–drug interactions	19	1	1 (5)
		2	2 (11)
		3	12 (63)
		5	4 (21)

**Table 3 jcm-11-01154-t003:** Interactions between cannabis and drugs.

Drug	Severity	Probability	Mechanism	Comments and Recommendations ^a^
LEVEL 1				
Warfarin [12,13,14,15]	Severe	Probable	CYP2CP inhibition	No clinical trials were found; nevertheless, 4 case reports were found, where the INR, whose therapeutic range was between 2 and 3, increased to 6.9 ^b^, 4.6 ^b^, 7.2 ^b^, 11.6 ^b^ with the concomitant administration of CBD in the first case and with inhaled cannabis (recreational) in the three other cases. Symptoms such as gastrointestinal bleeding were observed. The use of cannabis with warfarin could be contraindicated. If necessary, adjust warfarin dosage and closely monitor the patient’s INR. Advise the patient to avoid using cannabis recreationally.
LEVEL 2				
Buprenorphine [16]	Moderate	Probable	CYP3A4 inhibition	A retrospective analysis with 32 patients reported concentrations of buprenorphine 170% ^b^ higher for those who consume cannabis (recreationally) concomitant with buprenorphine. In addition, in one case report, a patient experiment reported a 95% ^c^ decrease in serum levels of buprenorphine when stopping the use of cannabis. Avoid the use of buprenorphine with cannabis. If the combination is necessary, adjust buprenorphine dosage and monitor plasma levels of buprenorphine. Advise the patient to avoid using cannabis recreationally.
Tacrolimus [17,18,19]	Moderate	Probable	CYP3A4 inhibition	Two case reports and one clinical trial with 6 persons, where only one patient showed changes, with reported increases of 358% ^c^, 200% ^b^, and 77% ^c^ in the plasmatic level of this immunosuppressant with the use of CBD. Avoid the use of tacrolimus with cannabis. If the combination is necessary, adjust tacrolimus dosage and monitor plasma levels of tacrolimus. Individualize usage of CBD and assess each case before dosing.
LEVEL 3				
Clozapine [20]	Moderate	Possible	CYP1A2 induction	In one case report, a patient stopped the consumption of cannabis and cigarettes, and the plasma levels of clozapine increased by 230% ^c^. During this increase, the patient hallucinated. Adjust clozapine dosage and, if it is possible, monitor plasma levels.
Methadone [21]	Moderate	Possible	CYP3A4 and CYP2C19 inhibition	A case report evidences the administration of CBD oil to a patient having methadone treatment. Methadone levels increased by 117% ^c^, and somnolence and fatigue were reported. Adjust methadone dosage and, if it is possible, monitor plasma levels.
Clobazam [22,23,24]	Minor	Defined	CYP2C19 inhibition	In 3 clinical trials, clobazam concentration increased by 25% ^c^, 60% ^b^, and 20% ^b^ in patients receiving different doses of clobazam and CBD. In the 3 studies, the antiepileptic doses were reduced when it was necessary to reduce the adverse events. Adjust clobazam dosage and, if it is possible, monitor plasma levels.
Chlorpromazine [25]	Minor	Defined	Possible CYP1A2 induction	A clinical trial showed an increase of 50% ^b^ in clearance in subjects who consume recreational cannabis. Adjust chlorpromazine dosage and, if it is possible, monitor plasma levels.
Eslicarbazepine [22,26]	Minor	Defined	Unknown. Perhaps the delivery vehicle (sesame oil) in this formulation of CBD could contribute to this interaction	In a clinical trial, with a concomitant administration of CBD (Epidiolex), an increase of 24% ^c^ was evidenced. This change was statistically significant, but it was evaluated in only 4 subjects. Adjust eslicarbazepine dosage and, if it is possible, monitor plasma levels.
Hexobarbital [27]	Minor	Defined	Possible CYP3A4 inhibition	In a clinical trial, hexobarbital clearance was 35% ^c^ lower when CBD was administered, compared to when it was not administrated, in subjects who consume recreational cannabis regularly. Adjust hexobarbital dosage and, if it is possible, monitor plasma levels.
Indinavir [28,29,30]	Minor	Defined	Possible CYP3A4 inhibition	In a clinical trial, the maximum concentration of indinavir decreased by 14.1% ^b^ using THC cigarettes in patients who use indinavir. There were no statistically significant changes with dronabinol usage. Adjust indinavir dosage and, if it is possible, monitor plasma levels.
Ketoconazole [31,32]	Minor	Defined	CYP3A4 inhibition	In a clinical trial, the concomitant administration of ketoconazole with Sativex increased the AUC of THC by 82% ^c^ and the AUC of CBD by 84% ^c^. Adjust ketoconazole dosage and, if it is possible, monitor plasma levels.
Rifampicin [31,32]	Minor	Defined	CYP3A4 induction	In a clinical trial, the concomitant administration of rifampicin with Sativex decreased the AUC of THC by 24% ^c^ and the AUC of CBD by 84% ^c^. Adjust rifampicin dosage and, if it is possible, monitor plasma levels.
Stiripentol [33,34,35]	Minor	Defined	CYP2C19 inhibition	In a phase 1 study, stiripentol AUC increased by 60% ^b^ with the concomitant administration of CBD (Epidiolex). In another phase 2 study, the increase was 30% ^b^. Adjust stiripentol dosage and, if it is possible, monitor plasma levels.
Theophylline [36,37,38,39]	Minor	Defined	CYP1A2 induction	There is evidence from two clinical trials and one retrospective study. In these, subjects used recreational cannabis and smoked cigarettes. Clearance was calculated, and it increased by 40% ^b^, 42% ^c^, and 48% ^c^. Adjust theophylline dosage and, if it is possible, monitor plasma levels.
Valproate [22,40]	Minor	Defined	Possible UGT1A9 y UGT2B7 inhibition	In a clinical trial, after increasing the CBD (Epidiolex) dose, the valproate level did not change. Nevertheless, an increase in ALT and AST levels by 49% ^c^ and 55% ^c^, respectively, was noted. In another clinical trial, 39% ^c^ of the patients taking CBD and valproate developed thrombocytopenia, but the results were not statistically significant. Assess liver function before starting CBD and monitor liver function.
LEVEL 5				
Rufinamide [22]	Lack	Defined		Co-administration with CBD (Epidiolex) does not lead to significant changes in rufinamide levels.
Topiramate [22]	Lack	Defined		Co-administration with CBD (Epidiolex) does not lead to significant changes in topiramate levels.
Zonisamide [22]	Lack	Defined		Co-administration with CBD (Epidiolex) does not lead to significant changes in zonisamide levels.
Nelfinavir [28,29,30]	Lack	Defined		Co-administration with THC cigarettes does not lead to significant changes in nelfinavir levels and AUC.

^a^ The recommendations for the management of each interaction are according to the level of the clinical relevance of cannabis–drug interactions [8,9,10]. In addition, some information stated by the authors was included. ^b^ The percentage of change was presented explicitly in the article. ^c^ The percentages of change was calculated with data contains within articles. Abbreviations: INR, international normalized ratio; CBD, cannabidiol; THC, tetrahydrocannabinol; AST, aspartate aminotransferase; ALT, alanine aminotransferase.

## Data Availability

Data is contained within this article. In addition, data in Table 3 with footnote “c” the percentages of change was calculated with data contains within articles.

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
