# Peer review of "Clinical Relevance of Drug Interactions with Cannabis: A Systematic Review"

_jcm, 2022, doi:10.3390/jcm11051154_

Round 1
Reviewer 1 Report
This is a review of the drug interactions of the different products derived from cannabis for both recreational and medicinal use.
Methods.
- I don't understand why you don't refer to your review as a systematic review instead of “structured review” as you have written. To follow the PRISMA standards, perhaps, it would be a good election.
- Exclusion criteria include articles that were not available in full text. It is not clear; it seems that this condition limits the search to only articles published in Open Access journals. This seems like a limitation that could be avoided by using the library of a university or some institutional system of shared payment access.
- The wording of line 86 should be changed. You have previously defined that one of the classification criteria will be severity. Maybe it should be written as follows: To determine severity, the magnitude of the qualitative change was taken into account .......
- To determine the severity, you deeply explain the method followed. However, you do not explain the method followed to calculate the probability. You say that it will be established based on the number of cases found and the type of article where the evidence is found. According to the information available in Table 2, it seems that you have followed a certain criterion in such a way that you establish as “defined” those interactions that come from clinical trials and as “probable or possible” those that come from observational studies or cases. This criterion or this decision algorithm It is not defined in the methods section and must be defined there appropriately.
- Additionally, you do not define how you did to classify the interactions in the five levels. It seems clear that each interaction will correspond to a level depending on its severity and probability, but how did you combine these criteria must be made explicit in the method section.
Discussion.
- On line 165, parentheses, you repeat data from the results that should not appear in the discussion.
Tables.
- In table 1, in the last column, it would appreciate the N to appears, that is, the number of interactions that correspond to each level. The percentage is shown, and the number of interactions could be calculated in absolute value, but let it be explicitly stated.
- In Table 2, the recommendations for the management of each interaction are not clear if they come from the recommendations of the authors themselves in the original articles, or if they are recommendations left to you at the time of making this review. A little explanation on this matter would be convenient at the bottom of the table.
- In Table 2, in the last column, the type of cannabis (drugs, recreational, etc) that was evaluated in each of the studies is adequately described, which is very interesting. However, in the line corresponding to buprenorphine, this information does not appear. A general reference to “cannabis”, without indicating whether they were commercial pharmaceutical products or for recreational use, is shown.
References.
- It strikes me that the following paper has not been cited: Schaiquevich P, Riva N, Maldonado C, Vázquez M, Cáceres-Guido P. Clinical pharmacology of cannabidiol in refractory epilepsies. Farm Hosp. 2020; 44 (5): 222-9. It includes a great review of the interactions of cannabidiol with antiepileptic drugs.
Author Response
Methods.
1. I don't understand why you don't refer to your review as a systematic review instead of a “structured review” as you have written. To follow the PRISMA standards, perhaps, it would be a good election.
Thank you for your suggestion. We agree that it is a systematic review; however, the results of this systematic review are qualitatively summarized; thus, it is a narrative systematic review. According to your suggestion, we have added it to the title “systematic review”. In addition, both in the abstract and in the main text we have indicated that it is a narrative systematic review. In addition, in methods we have included the following sentence: “The studies identified were reviewed by two researches in a paired way according to the Preferred Reporting Items for Systematic Reviews and Meta-Analysis (PRISMA) flow chart via pre-determined eligibility criteria”.
2. Exclusion criteria include articles that were not available in full text. It is not clear; it seems that this condition limits the search to only articles published in Open Access journals. This seems like a limitation that could be avoided by using the library of a university or some institutional system of shared payment access.
Thank you for your commentary. In methods, we have precise that criteria was used, therefore we have changed “articles without full-text access” by “Articles without full-text available”. We have access to all articles with full text available using the University of Antioquia system or asking the author for the pdf of his/her paper. As consequence, in the identification stage, only one article (of 94) was excluded (because the abstract and full-text were not available); and in the screening stage, only one article (of 53) was excluded because the full-text was not available. According to your suggestion, the Preferred Reporting Items for Systematic Review and Meta-Analysis (PRISMA) flow diagram was adjusted (figure 1).
3. The wording of line 86 should be changed. You have previously defined that one of the classification criteria will be severity. Maybe it should be written as follows: To determine severity, the magnitude of the qualitative change was taken into account .......
Thank you for your suggestion. Therefore, to improve the compression we have precise the probability and severity of the variable as followed: The severity was determined according to the magnitude of the qualitative change in parameters associated with efficacy or safety of the drug, such as pharmacokinetics (clearance or area under the curve–AUC-) or clinic (aspartate aminotransferase –AST- alanine aminotransferase-ALT- or International Normalized Ratio -INR- levels).
4. To determine the severity, you deeply explain the method followed. However, you do not explain the method followed to calculate the probability. You say that it will be established based on the number of cases found and the type of article where the evidence is found. According to the information available in Table 2, it seems that you have followed a certain criterion in such a way that you establish as “defined” those interactions that come from clinical trials and as “probable or possible” those that come from observational studies or cases. This criterion or this decision algorithm It is not defined in the Methods section and must be defined there appropriately.
Thank you for your suggestion. You right, since 2007 we published a proposal with this details (reference 8 - Amariles P, Andrés Giraldo N, José Faus M. Interacciones medicamentosas: aproximación para establecer y evaluar su relevancia clínica. Med Clin (Barc) 2007; 129: 27–35).
Therefore, according to your suggestion, in methods, we have detailed this topic.
The probability was determined and classified according to clinical support (type of study that supported the interaction) found for each pair of cannabis drug-interaction:
- Possible: The Interaction was documented by results from less than three case reports.
- Probable: The Interaction was documented by results from a least one observational study (cohort or case-control study) or a least three case reports.
- Defined: The Interaction was documented by results from a least one me-ta-analysis, narrative systematic review, or randomized or non-randomized clinical trial.
5. Additionally, you do not define how you did to classify the interactions in the five levels. It seems clear that each interaction will correspond to a level depending on its severity and probability, but how did you combine these criteria must be made explicit in the method section.
Thank you for your suggestion. According to your suggestion, in methods we have added a table with this topic, which is based in references 8 (above mentioned) and 10 (Amariles P, Madrigal-Cadavid J, Giraldo NA. Relevancia clínica de las interacciones medicamentosas: Propuesta de actualización de la clasificación, acorde con su gravedad y probabilidad. Rev Chilena Infectol. 2021; 38: 304-305)
In addition to the 4 levels of relevance proposed in 2007, in 2021 we proposed to include a level 5 of clinical relevance. This level 5 is characterized by evidence of lack of effect on the patients’ health (absence of severity of the interaction), which is documented by results from meta-analyses, systematic reviews, or clinical trials (defined probability) and, therefore, with evidence of absence of clinically relevant drug-interaction. Therefore, this level 5 corresponds to pair of drugs interactions with evidence that their simultaneous use does not generate clinically relevant interactions.
Therefore, in discussion we have added the following paragraph: Four pairs of cannabis-drug interactions were classified as level 5 of clinical relevance (interactions with evidence of lack of effect in patient´s health). This information is relevant for clinicians, because identifies combinations of drug with evidence that there are no risk for patients by drug interactions [10].
Discussion.
1. On line 165, parentheses, you repeat data from the results that should not appear in the discussion.
Thank you for your commentary. We repeat this data, because we believe that these figures are the most important results from this review. As consequence, we considered that could be necessary repeat these results in discussion.
Tables.
- In Table 1, in the last column, it would appreciate the N to appear, that is, the number of interactions that correspond to each level. The percentage is shown, and the number of interactions could be calculated in absolute value, but let it be explicitly stated.
Thank you for your commentary. Table 1 (current 2) was adjusted according to your suggestion.
Table 2. Pharmacological group and level of relevance of 19 pairs of cannabis-drug interactions identified
Pharmacological group |
Number of interactions |
Level of relevance |
Number by the level of relevance (%) |
Nervous system |
13 |
2 |
1 (8) |
3 |
9 (69) |
||
5 |
3 (23) |
||
Blood and hematopoietic organs |
1 |
1 |
1 (100) |
Anti-infectives for systemic use |
4 |
3 |
3 (75) |
5 |
1 (25) |
||
antineoplastic agents and immunomodulators |
1 |
2 |
1 (100) |
Total of pairs of cannabis-drug interactions |
19 |
1 |
1 (5) |
2 |
2 (11) |
||
3 |
12 (63) |
||
5 |
4 (21) |
2. In Table 2, the recommendations for the management of each interaction are not clear if they come from the recommendations of the authors themselves in the original articles, or if they are recommendations left to you at the time of making this review. A little explanation on this matter would be convenient at the bottom of the table.
Thank you. According to your suggestion, the following text was added: “The recommendations for the management of each interaction are according to the level of the clinical relevance of cannabis-drug interaction [8-10]. In addition, some information stated by the authors was included.”
In addition, due to the inclusion of another footnote, * and ** were changed by b and c. Therefore, current footnotes are a, b and c.
3. In Table 2, in the last column, the type of cannabis (drugs, recreational, etc) that was evaluated in each of the studies is adequately described, which is very interesting. However, in the line corresponding to buprenorphine, this information does not appear. A general reference to “cannabis”, without indicating whether they were commercial pharmaceutical products or for recreational use, is shown.
Thank you. According to your suggestion, the information regarding Buprenorphine, (recreational) was added
References.
- It strikes me that the following paper has not been cited: Schaiquevich P, Riva N, Maldonado C, Vázquez M, Cáceres-Guido P. Clinical pharmacology of cannabidiol in refractory epilepsies. Farm Hosp. 2020; 44 (5): 222-9. It includes a great review of the interactions of cannabidiol with antiepileptic drugs.
Thank you for the commentary. Using the strategy search, this article was not identified. However, it is a reinforcement for a commentary included about cannabis-clobazam interaction (For example, for clobazam and CBD we found three clinical trials, representing the cannabis-drug interaction with more evidence in this review). Thus, the reference was added (reference 45), after this text: “Similarly, one review oriented to systematize and update the CBD pharmacology in the context of refractory epilepsy founded that the strongest evidence is for cannabis-clobazam interaction [45].).”

Reviewer 2 Report
Comments and suggestions are attached.

Author Response
Dear editor and reviewers:
Subject: Comments article clinical relevance of cannabis-drug interactions
Thank you for your commentaries and suggestions, which have notoriously contributed to improve the quality and value of our work.
We have answered all commentaries in this document, and we I have did the changes in the article text, which was indicated in green color.
Reviewer 2.
- I would suggest including other inflammatory disorders and then discussing the mechanism of how this is proposed to work. The evidence is not robust for inflammatory bowel disorders as an example: https://pubmed.ncbi.nlm.nih.gov/31613959/.
Thank you for your suggestion. We have added the following text: “However, the evidence is not robust for COVID-19 and for another inflammatory disease such as Crohn's disease or ulcerative colitis” using the recommended reference (Reference 4). We consider that this information is adequate because it makes the information more accurate.
- There are more citations that should be used to support this and several studies available.
Thank you for your suggestion. We agree that there are several references that support this affirmation; however, consider that this text are oriented to show that cannabis usage is increasing with this citation is enough.
- There are 3 including Epidiolex (cannabidiol).
Thank you for the suggestion. We have added the following text: “and Epidiolex, which is an oral solution of CBD.
- Need to discuss potency and lack of standardization in measurement and testing as well as lack of dosing recommendations and discuss how recreational and medical differ.
Thank you for your recommendation. We agree that the lack of standardization in measurement and testing as well as lack of dosing recommendations for cannabis are limitations (we included this as a limitation of our work). In addition, we have included the following text in the discussion section:
Seven pairs of interactions (37%) were due to recreational use of cannabis, including one with warfarin. Thus, it is important to denote that the effect of drug interactions in the patient´s health with recreational cannabis can have a wide range according to the ingestion via, because it affects the type and quantity of cannabinoids absorbed.
To improve the rational use of cannabis medicinal is necessary more research aimed to: a) generate information regarding drug interactions; b) to determinate the type and concentrations of cannabinoids in the respective dosage forms; and c) to improve the standardization the quantities and dosage recommendations.
So, in limitations we added: In addition, some studies did not describe the cannabis concentration or cannabinoids type used; therefore, similar as happen with the use of recreational cannabis, the effect of the cannabis-drug interaction in the patient´s health can have a wide range due to type and quantity of cannabinoids absorbed.
5. What about other terms such as cannabinoid or cannabidiol?
Thank you for your commentary. It is possible that after this general review regarding cannabis, the inclusion of these terms in the search strategy generates information that is more detailed. Thus, now we are planned to conduct a systematic review adding these terms.
- Spelling of Addjust.
Thank you. The spelling of this word and to others were corrected.
- Buprenorphine - What type of study?
Thank you for your commentary. According to the type of study was included “A retrospective analysis with 32 patients” …
- So there is no level 4?
Thank you for your commentary. The information and evidence regarding cannabis-drug interactions are limited; with the information identified through this review drug interactions classified as level 4 of relevance (Minor and possible) were not founded.
- But the evidence was not ranked and is weak. That limits application.
Thank you for your commentary. We agree, thus we included this in limitations. However, for the 19 pairs of drug interactions identified this information could be used for this proposal.
- Spelling adviced.
Thank you for your commentary. The spelling of this word and to others were corrected.
- This needs to be updated to indicate that it is now FDA approved for two types of seizures in US (not all seizures)
Thank you for your commentary. According to the information found, in the discussion section, we update the information. “This result may be due to the fact that CBD is a therapeutic option as an anticonvulsant for rare epileptic disorders as Lennox-Gastaut Syndrome and Dravet Syndrome [22], which would motivate the research for interactions between cannabis with conventional antiepileptic drugs.”
- This sentence seems incomplete.
Thank you for your commentary. According to the comment, we have adjusted the sentence and the entire paragraph: In general, the evidence of cannabis-drug interactions continues to be limited. However, for some drugs this subject is more precise. For example, for clobazam and CBD we found three clinical trials, representing the cannabis-drug interaction with more evidence in this review. Similarly, one review oriented to systematize and update the CBD pharmacology in the context of refractory epilepsy founded that the strongest evidence is for cannabis-clobazam interaction [45].
- I would clarify that route of administration and other individualized factors (including prior use of cannabis) impact this. Smoke is different. What about vaporization? How about additives and solvents or other ingredients?
Thank you for your suggestion. We added more information regarding the different effects that via of administration can have (see commentary 4)…In addition, we detailed how smoking cigarettes can affect the interaction between a drug and smoking cannabis.
- That is a huge limitation. They also likely mixed doses.
Thank you for your commentary. We agree is an important limitation, thus the importance to generate information regarding this situation.

Reviewer 3 Report
This manuscript deals with cannabis related drug interactions with special emphasis on the clinical relevance. The authors searched systematically for data in the literature and evaluated the reported information in a well structured way. At the end, drug pairs were classified. Three pairs were considered as severe or moderate. The analysis was well done and the publication was well written. I have only two points for revision.
- Abstract
In the abstract, the authors should be more concrete and point to the 3 pairs with warfarin, buprenorphine and tacrolimus (level 1 and 2) that were considered as clinically relevant. - Introduction
Pharmacokinetic drug interactions with cannabinoids highly depend on the presence of different constituents. Especially the presence of CBD seems likely to have inhibitory potential. This is mentioned in the introduction. Some more information should be given. I recommend to refer to the following two recent publications:
Narin et al. 2021. Cannabinoid metabolites as inhibitors of major hepatic CYP450 enzymes, ..... (PMID 34493602) and
Bansal et al. 2020 Predicting the potential for cannabinoids to precipitate pharmacokinetic drug interactions .... (PMID 32587099)
Author Response
Dear editor and reviewers:
Subject: Comments article clinical relevance of cannabis-drug interactions
Thank you for your commentaries and suggestions, which have notoriously contributed to improve the quality and value of our work.
We have answered all commentaries in this document, and we I have did the changes in the article text, which was indicated in green color.
Reviewer 3.
1. Abstract. In the abstract, the authors should be more concrete and point to the 3 pairs with warfarin, buprenorphine, and tacrolimus (level 1 and 2) that were considered clinically relevant.
Thank you for your suggestion. According to it, we have added to the abstract “Cannabis-Warfarin was classified in level 1, cannabis-Buprenorphine, and tacrolimus in level 2”
- Pharmacokinetic drug interactions with cannabinoids highly depend on the presence of different constituents. Especially the presence of CBD seems likely to have inhibitory potential. This is mentioned in the introduction. Some more information should be given. I recommend referring to the following two recent publications:
Narin et al. 2021. Cannabinoid metabolites as inhibitors of major hepatic CYP450 enzymes, ..(PMID 34493602) and
Bansal et al. 2020 Predicting the potential for cannabinoids to precipitate pharmacokinetic drug interactions .... (PMID 32587099)
Thank you for your suggestion. We have added to the text “Cannabinoids, especially CBD, are metabolized by isoenzymes of the family CYP, which can lead to interactions between a Cannabis base extract with other drugs. In this way, recent in- vitro studies have shown that: THC competitively inhibits the enzymes CYP1A2, CYP2B6, CYP2C9, and CYP2D6; CBD competitively inhibits the enzymes CYP3A4, CYP2B6, CYP2C9, CYP2D6, and CYP2E1; and cannabinol (CBN) competitively inhibits the enzymes CYP2B6, CYP2C9, and CYP2E1 [9].”, citing the first article you suggest to us (Reference 9)

Reviewer 4 Report
Very good review of clinically significant interactions.
I would like to point out that there are two other possibly significant interactions which should be included:
1) clopidogrel. The following review is based on preclinical data, which is part of your exclusion criteria, but worth mentioning due to the potential clinical impact of this interaction (reduced conversion to active metabolite): Greger, J., Bates, V., Mechtler, L. & Gengo, F. A Review of Cannabis and Interactions With Anticoagulant and Antiplatelet Agents. J Clin Pharmacol 60, 432–438 (2020).
2) checkpoint inhibitors. A clinical study has demonstrated reduced treatment response of nivolumab with concomittant use of cannabis, though not CYP450 related. This is also clinically relevant, considering the increased use of these agents: Taha, T. et al. Cannabis Impacts Tumor Response Rate to Nivolumab in Patients with Advanced Malignancies. Oncologist (2019) doi:10.1634/theoncologist.2018-0383.
Both of these interactions may have an impact on survival and should be included in this review, in my opinion.
Author Response
Dear editor and reviewers:
Subject: Comments article clinical relevance of cannabis-drug interactions
Thank you for your commentaries and suggestions, which have notoriously contributed to improve the quality and value of our work.
We have answered all commentaries in this document, and we I have did the changes in the article text, which was indicated in green color.
Reviewer 4.
Comments and Suggestions for Authors
Very good review of clinically significant interactions. Thank you for your commentary.
I would like to point out that there are two other possibly significant interactions which should be included:
- The following review is based on preclinical data, which is part of your exclusion criteria, but worth mentioning due to the potential clinical impact of this interaction (reduced conversion to active metabolite): Greger, J., Bates, V., Mechtler, L. & Gengo, F. A Review of Cannabis and Interactions With Anticoagulant and Antiplatelet Agents. J Clin Pharmacol60, 432–438 (2020).
Thank you for your suggestion. Using the search strategy, this article was not identified; however, we consider that is important to emphasize in the evidence of warfarin interactions because this interaction is assessed by our review as the one with the highest risk. Therefore, we have considered this article to be included in the discussion Reference 41) and we have added to the text: “In addition, Greger et al, [41] in a review of warfarin interactions founded in-vivo evidence (case reports) regarding cannabis-warfarin interaction, and theorized other interactions, for example, with clopidogrel, based on the isoenzymes that metabolized this drug and cannabis.
- checkpoint inhibitors. A clinical study has demonstrated reduced treatment response of nivolumab with concomitant use of cannabis, though not CYP450 related. This is also clinically relevant, considering the increased use of these agents: Taha, T. et al.Cannabis Impacts Tumor Response Rate to Nivolumab in Patients with Advanced Malignancies. Oncologist (2019) doi:10.1634/theoncologist.2018-0383.
Thank you for your suggestion. Using the search strategy, this article was not identified; however, this reference is useful to give one example of drug interaction with still limited evidence of the effects that can cause. Therefore, we have considered this article to be included in the discussion (Reference 46), and we have added to the text “In contrast, there many theorized cannabis-drug interactions or with clinical evidence limited. For instance, the interaction between cannabis products (for smoking or oils) in patients treated with nivolumab. In a retrospective study the use of cannabis were associated to minor response rate in patients in immunotherapy treatment with nivolumab. However, the magnitude in the change in response rate was not relate to cannabis composition. Therefore, to determine the mechanism and clinical relevance of this interaction it is necessary to conduct clinical trials and generates more information and evidence of this cannabis-nivolumab interaction [46].

Round 2
Reviewer 2 Report
I am satisfied with the changes to this manuscript and the rationale.
Reviewer 4 Report
Corrections are appropriate. Good work.